# The Influence of the Degree of Thermal Inactivation of Probiotic Lactic Acid Bacteria and Their Postbiotics on Aggregation and Adhesion Inhibition of Selected Pathogens

**DOI:** 10.3390/pathogens11111260

**Published:** 2022-10-29

**Authors:** Marcelina Karbowiak, Michał Gałek, Aleksandra Szydłowska, Dorota Zielińska

**Affiliations:** Department of Food Gastronomy and Food Hygiene, Institute of Human Nutrition Sciences, Warsaw University of Life Sciences (WULS-SGGW), 02-776 Warsaw, Poland

**Keywords:** lactic acid bacteria, heat-killed bacteria, postbiotics, adhesion, co-aggregation, auto-aggregation, hydrophobicity

## Abstract

The study aimed to evaluate the effect of thermal inactivation of potentially probiotic lactic acid bacteria (LAB) strains isolated from food on their ability to compete with pathogenic microorganisms. Five strains of LAB, previously isolated from food and characterized, one commercial reference strain of *Lactiplantibacillus plantarum* 299v, and two indicator strains of *Staphylococcus aureus* 25923 and *Listeria*
*monocytogenes* 15313 were used in the study. The experiment consisted in applying a stress factor (high temperature: 80 °C, at a different time: 5, 15, and 30 min) to the tested LAB cells to investigate the in vitro properties such as hydrophobicity abilities (against *p*-xylene and *n*-hexadecane), auto-aggregation, co-aggregation with pathogens, and inhibition of pathogens adhesion to the porcine gastric mucin. The bacterial strains showed various hydrophobicity to *p*-xylene (36–73%) and *n*-hexadecane (11–25%). The affinity for solvents expanded with increasing thermal inactivation time. All LAB isolates were able to auto-aggregate (ranging from 17 to 49%). Bacterial strains subjected to 5 and 15 min of thermal inactivation had the highest auto-aggregation ability in comparison to viable and heat-killed cells for 30 min. The LAB strains co-aggregated with pathogens to different degrees; among them, the highest scores of co-aggregation were observed for *L. monocytogenes*, reaching 27% (with 15 min of heat-killed LAB cells). All LAB strains reduced the adherence of pathogenic bacteria in the competition test, moreover, heat-killed cells (especially 15 min inactivated) were more efficient than viable cells. The properties of selected LAB strains as moderately heat-stressed forms analyzed in the study increased the prevention of colonization and elimination of pathogenic bacteria in the in vitro model of gastrointestinal tract. The thermal inactivation process may therefore preserve and modifies some characteristics of bacterial cells.

## 1. Introduction

Despite the undeniable and significant advances in the food industry, the incidence of foodborne diseases is still on the rise in the European Union. In the year 2019, campylobacteriosis and salmonellosis were the most common zoonoses, and the highest number of deaths (of 31) was reported for listeriosis [1]. As a result, raising awareness of the risks associated with *L. monocytogenes* is especially important among high-risk consumer groups. However, most consumers today are highly opposed to food additives, considering them harmful, even if they are unable to explain how they affect their health [2] The goal of recent research has been to develop products that use fewer additives and/or natural ingredients, as well as to meet the consumer’s demands while maintaining food safety and quality [3]. In this context, natural antimicrobial agents have gained significant attention. The most promising biopreservatives include antimicrobial proteins, bacteriophages, probiotics, non-viable bacterial forms, and plant-based substances [4]. Among the most studied natural antimicrobial compounds are those produced by microorganisms, such as lactic acid bacteria (LAB). LAB constitute a broad heterogeneous group of generally food-born microorganisms used in food preservation. As a result of their health benefits, associated with their continuous consumption and colonization of the gut microbiota [5], they are often considered probiotics. The most common bacterial genus used as probiotics including *Lacticaseibacillus*, *Lactiplantibacillus*, *Levilactobacillus*, and many others, which were historically grouped within one genus named *Lactobacillus* [6]. According to the International Scientific Association for Probiotics and Prebiotics (ISAPP) consensus statement probiotics are “live microorganisms that, when administered in adequate amounts, confer a health benefit on the host” [7]. The application of probiotics into food has many advantages, including their interaction with pathogens, as probiotics may reduce pathogen growth by competing for nutrients or by secreting antimicrobial substances [8]. Although probiotics have several health benefits due to their ability to modulate the microbiome in the gut, all drawbacks related to the administration of viable microorganisms, have prevented their full potential for use in food and pharmaceutical applications. Few reports indicated that the administration of live microorganisms might be hazardous [9,10,11,12,13,14]. On the other hand, heat-killed preparations of microorganisms, including probiotics, and/or their preformed metabolites have recently gained attention, pointing out that there are some health benefits associated with physiologically active bacteria that are not directly related to their viability [15,16]. The terms “postbiotics” or “paraprobiotics” have been used to describe non-viable microorganisms or bacterial-free extracts that may provide additional benefits to the host through their bioactive qualities. Until recently, these two terms were treated as separate concepts. Postbiotics were defined as soluble cell-free supernatants containing products secreted by probiotics that have physiological benefits to the host [17]. While, the paraprobiotics were described as the inactivated microbial cells of probiotics, intact or ruptured containing cell components that confer a health benefit to the host [18]. However, the ISAPP’s (2021) definition of postbiotics brought some standardization in this matter in view of the “benefit from coalescing around the use of a single, well-defined and understood term rather than the use of disparate terms for similar concepts”, suggesting the term “postbiotic” defined as “preparation of inanimate microorganisms and/or their components that confers a health benefit on the host” [15]. Therefore, inactivated cells of probiotic strains that have demonstrated health benefits have fallen into this definition of postbiotics. So far, postbiotics are increasingly being used in food and pharmaceutical industries, and several postbiotic products derived from LAB species are available for use in the prevention or treatment of disease [19,20,21,22]. Even so, more evidence is necessary to validate postbiotics’ beneficial effects.

The use of postbiotics has several advantages over living organisms. The main argument for the safety of using inactivated cells of probiotic strains is the fact that such cells have lost their ability to proliferate [23]. Other beneficial properties of postbiotics include (1) accessibility in their pure form; (2) simplicity of production and storage; (3) particular mechanism of action; (4) no risk of bacterial translocation from the gut lumen to the blood in vulnerable or immunocompromised individuals; (5) no possibility of antibiotic resistance genes being acquired and transferred; and (6) more natural extraction and standardization [17,18,24,25]. Furthermore, in spite of the need for further investigation of the molecular mechanisms underlying postbiotic action, scientific evidence indicates that similarly to probiotics, molecules present on cell surfaces (peptidoglycan, teichoic acid, cell wall polysaccharides, cell surface-associated proteins, etc.) may represent the first line of interaction between postbiotics and the host, especially in case of antimicrobial activity, thereby promoting their beneficial effects [24]. Several reports indicated that postbiotics produced from LAB have exhibited broad antagonistic activities, demonstrating their potential to inhibit pathogens of various species [26,27,28,29]. Although the antimicrobial properties of LAB are obvious, the mechanism of action of their postbiotics, and the influence of the degree of LAB cell damage on this effect is still unclear.

Therefore, the purpose of the presented study was to evaluate in vitro the five potentially probiotic LAB strains previously isolated from food as viable, and heat-killed forms, against two model pathogen microorganisms: *Listeria monocytogenes* and *Staphylococcus aureus*. In addition, verification of how the heating time of LAB cells influences their selected biological properties that are important for adhesion capacity was investigated. To the best of the knowledge, this is the first study in which different heat-inactivation time of LAB strains was compared in terms of preserving their antimicrobial properties. The research is part of the current approach to the importance of providing more evidence to validate the beneficial effects of postbiotics for the future development of prophylactic or therapeutic agents as well as functional food or food additives.

## 2. Materials and Methods

### 2.1. Bacterial Strains and Growth Conditions

Five different LAB strains were used: *Lacticaseibacillus casei* O12, *Lacticaseibacillus casei* O16, *Levilactobacillus brevis* O22, *Levilactobacillus brevis* O24, and *Lactiplantibacillus plantarum* O20. The strains were isolated from traditional fermented food and molecular typing and in vitro probiotic properties of the strains were previously described [30,31]. The well-studied commercial probiotic strain *Lactiplantibacillus plantarum* 299v was used as a reference strain. In this study, the strains were marked with the corresponding symbols: O12, O16, O22, O24, O20, and 299v, respectively.

All bacterial strains were kept in a mixture of de Man, Rogosa, and Sharpe (MRS) broth (LabM, Heywood, United Kingdom) (800 μL) and glycerol (200 μL) at −80 °C until further use. LAB strains were twice subcultured on MRS broth every 24 h before experimental use. A serial dilution of sterile peptone saline (Lab M, Heywood, United Kingdom) in MRS agar (Merck, Darmstadt, Germany) was conducted to determine the colony-forming units CFU/mL. Incubation was managed in anaerobic conditions for 48 h at 37 °C.

Indicator strains used for evaluation of antimicrobial properties were *Listeria monocytogenes* strain (ATCC^®^ 15313™), and *Staphylococcus aureus* strain (ATCC^®^ 25923™) from the collection of American Type Culture Collection (ATCC). The following selective media was used to enumerate: Palcam Agar Base (Lab M, Heywood, United Kingdom) for *Listeria*, and Baird-Parker Agar (Merck, Darmstadt, Germany) for *Staphylococcus*. Singe colonies had been transferred to Nutrient broth (Noack Group, Vienna, Austria). Cultivation was carried out at 37 °C for 24–48 h aerobically. 

### 2.2. Thermal Inactivation of Bacterial Strains

Thermal inactivation was used as a fixation method to preserve the bacterial surface proteins unchangeable, as well as to kill bacteria cells entirely. For this purpose, cultured bacteria were precipitated by centrifugation in 5000 × *g* for 10 min. The supernatant was then decanted and the remaining pellet was suspended in 10 mL of PBS (phosphate buffer saline) solution (Pol-Aura, Dywity, Poland). This action was repeated twice. Afterward, the pellet was resuspended in an appropriate volume of PBS solution and standardized to OD_600_ = 1.0 (2 × 10^8^ CFU/mL) in the studies of hydrophobicity, auto-, and co-aggregation. Cultures with an OD_600_ value of 0.5 were prepared for the study concerning the ability to inhibit the adhesion of pathogens. The inactivation step was applied by placing the obtained solutions in the water bath at 80 °C, and then the appropriate volumes of suspensions were taken after 5, 15, and 30 min. Viable cells of LAB were also used in the study.

### 2.3. Cell Surface Hydrophobicity

Bacterial cell surface hydrophobicity was assessed by measuring microbial adhesion to solvents (MATS) as described by Pérez et al. [32] with minor modifications. Thermal inactivated strains with OD_600_ = 1, in the amount of 3 mL, were transferred to separate tubes. The samples were cooled to room temperature, and then 1 mL of *n*-hexadecane solvent (Chempur, Piekary Śląskie, Poland) or 1 mL of *p*-xylene solvent (Chempur, Piekary Śląskie, Poland) was added separately to each of the six tested strains. After the solvent was added, each sample was vigorously vortexed for 20 s. The samples were incubated for 20 min at room temperature. After incubation, the two phases were separated, and the aqueous phase was carefully removed, in the amount of 1000 µL was applied to a 24-well plate, and its absorbance at 600 nm in a spectrophotometer (Molecular Devices, San Jose, CA, USA) was measured. The hydrophobicity percentage was calculated using the following formula:H (%) = [A0 − A/A0] × 100(1)
where A0 and A are the absorbance values of the aqueous phase before and after contact with *n*-hexadecane/*p*-xylene, respectively.

Data were expressed as a mean ± standard deviation which was derived from the hydrophobicity values of three independent experiments.

### 2.4. Auto-Aggregation Assay

The auto-aggregation assay for LAB strains was done according to the method of Kos et al. [33] with certain modifications. The suspensions of specific strains (50 µL) were applied to a 24-well plate, each well previously filled with a volume of 1950 µL of PBS solution. Optical density was measured in a spectrophotometer (Molecular Devices, San Jose, CA, USA). The absorbance of the upper suspension was measured at intervals (0, 1, and 24 h) without disturbing the microbial suspension, and the kinetics of sedimentation was obtained. The auto-aggregation coefficient was calculated at time t as:A (%) = [1 − (At/A0)] × 100(2)
where A0 is the initial optical density at 600 nm of the microbial suspension and At is the optical density at time t.

The auto-aggregation assay was also carried out for pathogen strains to acquire the auto-aggregation percentages so that the obtained results could be used to calculate the co-aggregation percentages.

Data were expressed as a mean ± standard deviation which was derived from the auto-aggregation values of six independent experiments.

### 2.5. Co-Aggregation Assays with Pathogens

Coaggregation was performed according to Kos et al. [33] with some modifications. An equal volume of each cell suspensions of co-aggregation partners (LAB strains and pathogens) were mixed. Samples were kept to stand for 24 h at room temperature. The absorbance of 600 nm of the bacterial suspensions was monitored at 0, 1, and 24 h, and the percentage of co-aggregation was expressed as follows:C (%) = {[(Ax + Ay)/2 − A(x + y)]/[(Ax + Ay)/2]} × 100(3)
where Ax and Ay represent the absorbance at 600 nm in control tubes containing only the pathogen or the LAB strain respectively and A(x + y) represents the absorbance at 600 nm of the mixed culture after different periods of incubation.

Data were expressed as a mean ± standard deviation which was derived from the co-aggregation values of four independent experiments.

### 2.6. Inhibition of Pathogens Adhesion

Adhesion assays (in vitro) were performed according to Moisés Laparra et al. [34] with some modifications. Mucin from the porcine stomach (MilliporeSigma, Burlington, MA, USA) at a concentration of 0.5 mg/mL PBS was applied in a volume of 200 µL per well of a 96-well microtiter polystyrene plate by incubation at 4 °C overnight. After this time, unbound mucin was drained and each well was rinsed three times with PBS solution. *L. monocytogenes* and *S. aureus* cell suspensions were centrifuged 5000 × *g* for 10 min. The resulting pellets of bacterial cells were subjected to fluorescence staining by dissolving the pellet in 1 mL of CFDA dye (5-carboxyfluorescein diacetate) (MilliporeSigma, Burlington, MA, USA) at a concentration of 100 µmol/L. The cultures were incubated for 30 min at 37 °C. After this time, the cultures were centrifuged and washed 2 times in 5 mL PBS. Then 100 μL of viable or heat-killed LAB strains together with stained cells of pathogens suspension were added to the wells coated with mucin and incubated for 1h at 37 °C. Thereafter, the wells were washed two times with PBS buffer and adhered cells were recovered by treating each well with 200 μL of 1% (w/v) SDS (sodium dodecyl sulfate) (Thermo Fisher Scientific, Waltham, MA, USA) in 0.1 M NaOH solution for 30 min at 37 °C. At this time, the contents of each well were transferred in the same order to a black 96-well fluorescence plate. The fluorescence was measured at λex = 488 nm and λem = 538 nm in a spectrophotometer (Molecular Devices, San Jose, CA, USA). Mucin with PBS addition without added bacteria was used as negative control and the absorbance value of this negative control was subtracted from the absorbance value of the samples.

To analyze the results of inhibition of pathogen adhesion, a calibration curve for *L. monocytogenes* and *S. aureus* was constructed to estimate the number of mucin-bound bacteria (CFU/mL), from which the percentage of pathogen cells reduced was then calculated. Serial dilutions of the contents of each well were made in PBS (1: 0, 1: 1, 1: 2, 1: 4, 1: 8, 1:16, and 1:32). The solutions were applied in an amount of 200 µL to a 96-well plate and at wavelengths identical to the previous experiment, fluorescence was measured and standard curves were prepared.

Percentage inhibition of pathogens adhesion was calculated as follows:IA (%) = [after adhesion assay (CFU/mL)/before adhesion assay (CFU/mL)] × 100(4)

Data were expressed as a mean ± standard deviation which was derived from the inhibition of pathogens adhesion values of six independent experiments.

### 2.7. Statistical Analysis

The STATISTICA 13.3 Software (StatSoft Inc., Tulsa, OK, USA) was used to perform the statistical analysis. All results (cell surface hydrophobicity, auto-aggregation, co-aggregation, and inhibition of pathogen adhesion to the mucin) obtained in the study were statistically analyzed using a two-way ANOVA. The model included the effect of time of the heat inactivation (t = 0, t = 5′, t = 10′, and t = 15′) of tested bacterial strains (*L. casei* O12, *L. casei* O16, *L. brevis* O22, *L. brevis* O24, *L. plantarum* O20, *L. plantarum* 299v) and their interaction (heat inactivation time x bacterial strains). In each of the studies, the effect of the heat inactivation time and the effect of the type of strain on potentially probiotic properties were subjected to a one-way analysis of variance (ANOVA). The comparison of *post-hoc* means was performed with Tukey’s range test. When the assumptions of a parametric test were not respected, Kruskal-Wallis, a nonparametric test, along with Dunn’s multiple comparison test was applied. *p* values less than 0.05 were considered statistically significant.

## 3. Results

### 3.1. Cell Surface Hydrophobicity

The isolates were tested for their cell surface hydrophobicity to estimate their adhesion ability, using the hydrocarbons *p*-xylene and *n*-hexadecane (Table 1).

The LAB isolates showed different hydrophobicities in terms of the affinity to *p*-xylene and *n*-hexadecane while subjected to thermal inactivation. Hydrophobic cell surface was shown by high adherence to *p*-xylene (36.03–72.97%) followed by *n*-hexadecane (11.00–24.73%).

A significant influence of heat inactivation time (*p* < 0.001) and type of bacterial strains (*p* < 0.001) on the cell surface hydrophobicity against *p*-xylene was observed. The least affinity to *p*-xylene was observed for viable cells of commercial strain *L. plantarum* 299v (36.03%). Nevertheless, a significant (*p* < 0.05) increase in hydrophobicity was observed between viable cells and heat-killed cells for 5 and 30 min in the case of *L. plantarum* 299v. In addition, a significant increase (*p* < 0.05) in the affinity of bacterial isolates for *p*-xylene regarding viable cells was observed for strains *L. brevis* O22 (for a 30-min inactivation), *L. casei* O16 (for 15- and 30-min inactivation), and *L. brevis* O24 (for 5-, 15- and 30-min inactivation).

Significant differences in the cell surface hydrophobicity of bacterial strains against *n*-hexadecane based on the heat inactivation time (*p* < 0.001) and type of bacterial strain (*p* < 0.001) were found. This effect was not seen in a combination of those two factors. When tested for cell surface properties using *n*-hexadecane, the highest hydrophobicity was determined for heat-killed cells for 30 min of the *L. brevis* O22 strain (24.80%). A significant increase of hydrophobicity (*p* < 0.05) in comparison to viable cells was detected for heat-killed cells (t = 15') of *L. casei* O16 strain (17.23%). In this case, a decrease in the hydrophobic properties of *L. plantarum* 299v cells was observed after exposing them to thermal stress. Anyhow, the result did not reach statistical significance.

### 3.2. Auto-Aggregation Assay

Six LAB strains exhibited auto-aggregation ranging from 17.08 to 48.87% after incubation at 37 °C, as shown in Figure 1. In this case, no significant differences were observed, both between the tested strains within different inactivation time or between the different heat inactivation time within the same strain, as well as after an hour of incubation and 24 h of incubation.

After an hour of incubation, the results indicated that the heat inactivation time and type of bacterial strains separately significantly affected the auto-aggregation properties (*p* < 0.001). The auto-aggregation values were in the range from 17.08% in 30 min heat-killed forms of commercial *L. plantarum* 299v strain to 38.59% in 15 min heat-killed forms of *L. plantarum* O20 strain. There was a trend that with one-hour incubation of bacterial cells, the greater ability to aggregate was demonstrated by cells that were quenched with high temperature for 5 min and 15 min (*p* > 0.05). On the other hand, treating bacteria cells with temperature for 30 min weakened their power to clump together (*p* > 0.05). The values of auto-aggregation that characterized all heat-killed forms for 30 min were lower than the control samples, except for the *L. brevis* O20 strain.

After 24 h of incubation, the values of auto-aggregation fluctuated in the range from 24.13 to 48.87%. In this case, heat-killed forms for 5 and 15 min were also characterized by the greatest ability to form auto-aggregates. Heating the cells for 30 min reduced their auto-aggregation potential. The strain *L. casei* O12 was the one with the highest percentages of auto-aggregation (as well as viable and all heat-killed forms). The weakest auto-aggregation ability was demonstrated by heat-killed forms for 30 min of *L. brevis* O22 and O24 strains.

### 3.3. Co-Aggregation with Listeria monocytogenes

The co-aggregation ability of investigated LAB with *L. monocytogenes cells* is shown in Figure 2. The bacterial mixtures were monitored after 0, 1, and 24 h of incubation.

All tested strains showed some co-aggregation properties with the foodborne pathogen of *Listeria monocytogenes*. At time 0, heat-killed forms for 15 min of the *L. casei* O12 strain showed the highest co-aggregating ability (27.03%) with *L. monocytogenes*, whereas the lowest was noticed in the case of commercial strain *L. plantarum* 299v viable cells (6.69%). A significant increase (*p* < 0.05) in co-aggregation ability between viable and heat-killed cells was observed solely in commercial strain *L. plantarum* 299v. In other strains, the percentage of co-aggregation ability fluctuated insignificantly. The results for this incubation time indicate that the ability to form co-aggregates was influenced only by the strain type (*p* < 0.05).

All treatments of the *L. casei* O12 strain exhibited the best co-aggregation properties after an hour of incubation, particularly 15 min heat-killed forms of the *L. casei* O12 strain showed the highest co-aggregation ability (12.72%), which was higher (*p* < 0.05) than the percentages of the co-aggregation of *L. plantarum* O20 strain.

All tested viable and heat-killed forms showed some co-aggregation ability with *Listeria monocytogenes* after 24 h of incubation. This ability was thermal inactivation time-specific (*p* < 0.05), and the percentages fluctuated. Significant differences in the co-aggregation ability between heat treatments within the two strains were noted. Co-aggregation of 15 min of heat-killed forms (12.74%) in comparison to 5 min of heat-killed forms (1.31%) and 30 min (1.40%) of the *L. casei* O16 strain was greater (*p* < 0.05). On the other hand, the ability of viable cells of *L. brevis* O22 strain (10.20%) to co-aggregate with the pathogen was greater (*p* < 0.05) than that of cells of the same strain but exposed to thermal shock for 5 min (1.13%) and 30 min (0.83%).

### 3.4. Co-Aggregation with Staphylococcus aureus

The co-aggregation between *Staphylococcus aureus* and viable and heat-killed forms of investigated LAB stains is indicated in Figure 3.

It can be observed that the ability to co-aggregate with *S. aureus* seems to be species-dependent (*p* < 0.001). *L. casei* O12 and O16 were the best aggregating strains, while *L. brevis* O22 and O24 were the worst. No unequivocal tendency was found in the case of the tested *L. plantarum* species. The probiotic strain *L. plantarum* 299v co-aggregated successfully, while *L. plantarum* O20 showed a poor ability to co-aggregate. In this case, a negative co-aggregation capacity between the strains of *L. plantarum* O20, *L. brevis* O22, and *L. brevis* O24, and the pathogen *S. aureus* was observed.

At time 0, significant differences were observed between viable and heat-killed forms, consisting of a significant increase in co-aggregation in favor of the latter. Co-aggregation of viable cells of *L. casei* O12 and *L. casei* O16 was 14.20 and 16.94% respectively, and co-aggregation of thermal inactivated cells for 15 min of corresponding strains amounted to 25.06 and 25.57% respectively.

Also, significant differences were found within viable and heat-killed cells of the *Lactobacillus* strain after 1-h co-aggregation. The co-aggregation ability of heat-killed cells for 5 and 15 min of the *L. casei* O12 strain (0.7%; 2.89%) was significantly lower than the coaggregation of viable cells (8.40%) and those heated for 30 min (8.85%). Similarly, the co-aggregation ability of viable cells of the *L. casei* O16 strain (8.23%) appeared substantially higher than the co-aggregation ability of all the bacterial heat-killed forms of this strain (0,79%; 0,72%; 1,01% respectively). A significant relationship (*p* < 0.05) was also observed between viable cells of the *L. brevis* O22 strain (7.08%) and those exposed to heat stress for 5 min (−14.57%).

The percentages of co-aggregation ability varied after incubation of cells for 24 h. A significant decrease (*p* < 0.05) in the co-aggregation ability percentage was reported only between heat-killed forms for 5 (19.51%) and 30 min (8.97%) of the *L. casei* O16 strain.

### 3.5. Inhibition of Pathogens Adhesion to the Mucin

In Table 2, the results of in vitro adhesion to the mucin studies are presented. All investigated LAB strains prevented the adhesion of the indicator microorganisms to the porcine gastric mucin layer (52.59–99.18%).

The results indicate that heat inactivation time (*p* < 0.001) and type of bacterial strain (*p* < 0.001) significantly affected the inhibition of pathogens’ adhesion to the mucin.

Significant differences (*p* < 0.05) between the inhibition of adhesion of *L. monocytogenes* to porcine gastric mucin by viable and heat-killed forms of the strains *L. brevis* O24 and *L. plantarum* O20 were noted. In the case of *L. casei* O16, *L. brevis* O22 strains, and commercial *L. plantarum* 299v strain, exposure of the cells to high temperature reduced the inhibition of pathogen adhesion to the porcine gastric mucin. The difference was significant (*p* < 0.05) for the *L. brevis* O22 strain.

It was also observed that *L. brevis* O24 and *L. casei* O16 heat-killed forms inhibited the adhesion of *S. aureus* to a greater extent (*p* < 0.05) in comparison to the control sample. This effect was seen when heat-killed cells for 30 or/and 15 min were used. The weakest reduction of adhesion by heat-killed cells was observed in the case of *L. brevis* O22—about 56.23%, compared to viable cells (*p* > 0.05).

## 4. Discussion

To inactivate bacterial cells and preserve their surface structures, different fixation methods have been demonstrated in the literature, such as inactivation by chemical agents, heat, sonication, and UV irradiations [35]. Each stress condition or agent used to inactivate microorganisms has its mechanism for inactivating the cell. There is no doubt that thermal inactivation is one of the most historical, and important preservation techniques [35]. Environmental stress can activate the cell defense mechanisms of microorganisms, resulting in an increased production of protein substances that are antagonistic to others bacteria, and therefore make it possible for these bacteria to survive in harsh conditions [36]. Cell surface properties such as hydrophobicity, auto-aggregation, co-aggregation, and adhesion inhibition could change under the influence of stress factors in comparison to viable cells in the case of probiotic bacteria. These are the phenomena that characterize the probiotic nature of the microorganism, which consists in attaching to various substrates [37]. However, there is now increased interest in research on the use of postbiotics, which constitute mainly metabolites of probiotics, or the other cell components [15]. In the present study, the effects of heat-inactivated six different bacteria were investigated. The thermal inactivation method was selected to inactivate the internal metabolism of bacteria and conserve their surface structures to find whether inactivated bacterial cells could have an effect on their ability to compete with pathogenic microorganisms or whether vitality is necessary for affecting.

### 4.1. Cell Surface Hydrophobicity

Cell hydrophobicity is an important attribute for probiotic bacteria, as well as for heat-inactivated cells due to its correlation to adhesion ability. The cell’s ability to adhere to the intestinal epithelium is considered a prerequisite for colonization of the human gastrointestinal tract so probiotics or postbiotics can exhibit a range of beneficial effects such as the exclusion of enteropathogenic bacteria [38,39]. It is suggested that bacteria with high hydrophobicity have a greater ability to bind to epithelial cells [40].

The adhesion of tested bacterial strains to solvent assay was conducted to investigate the features of the physicochemical cell surface (Table 1). The MATS (Microbial Adhesion to Solvent) technique was applied based on the affinity of microorganisms to monopolar and/or nonpolar solvents [41]. The bacterial adhesion to hydrocarbons has been extensively used for measuring cell surface hydrophobicity in lactic acid bacteria [38,40,42,43,44,45,46,47,48,49,50,51,52,53,54,55,56,57,58,59,60].

Adherence to a solvent *p*-xylene describes bacterial cell surface hydrophobicity. On the other hand, *n*-hexadecane was used as a nonpolar solvent having intermolecular attraction comparable to that of chloroform. The hydrophilic and hydrophobic features are attributed to polysaccharides and proteins on the bacterial surface [61]. Moreover, one other surface physicochemical characteristic of bacteria is electron donation and reception [61]. Therefore, the conducted study attempted to determine two surface characteristics for six different bacterial strains which represented viable cells or heat-killed cells for 5, 15, and 30 min.

In the present study, the cell surface hydrophobicity varied with the isolates tested. In the profuse majority of cases, the affinity for solvents expanded with increasing thermal inactivation time, and certainly the values for heat-killed cells were greater than those for viable cells with some significant differences (Table 1). Similar results of a thermal inactivation-dependent increase in hydrophobicity ability were obtained by others [45,56]. Shin et al. [56] noticed, that exposure of *Lactobacillus acidophilus* CBT LA1 cells to thermal stress at 80 °C for 10 min, increase the hexadecane hydrophobicity by 7.26%, and the difference was significant (*p* < 0.05) [56]. Collado et al. [45] have found that after heat stress (98°C for 10 min) of LAB cells the xylene hydrophobicity percentages increased from 9.6% to 17.8% for all tested bacteria [45]. Opposite results in the change of cell surface hydrophobicity were obtained by Markowicz et al. [23], where after heat treatment (80 °C for 20 min) the reduction in *n*-octane hydrophobicity of the *Lacticaseibacillus rhamnosus* GG cell surface was observed. In the conducted study the hydrophobic cell surface was shown by high adherence to *p*-xylene followed by *n*-hexadecane. High affinity to *p*-xylene represents an advantageous hydrophobic ability. On the other hand, a lower percentage of bacteria adhering to a nonpolar solvent such as *n*-hexadecane may be indicated that the tested strains had low hydrophobicity. Also, tested LAB strains still simultaneously showed some affinity to *p*-xylene and *n*-hexadecane hydrocarbons, suggesting a high complexity of the cell surface. The existing distinctions in the present study might be a consequence of varied surface molecules such as the presence of exopolysaccharides (EPS) and specific surface proteins, additionally influenced by the effect of high inactivation temperature.

### 4.2. Auto-Aggregation Assay

Bacteria are keen on existing in consortia, sticking together and/or to other bacterial cells, and adhering either to surfaces. Bacterial aggregation between cells of the same strain in a nonspecific way is called auto-aggregation. This process consists of the reaction of bacterial strains physically with each other and precipitates from static liquid suspension [62]. The ability of cells to form auto-aggregates appears to be another (after the cell surface hydrophobicity) essential prerequisite for the adhesion of bacterial cells to the intestinal epithelium. Probiotic bacteria ought to create adequately large biomass through auto-aggregation to succeed in the required advantage in the gastrointestinal tract. Moreover, auto-aggregation ability increases the chance of microbial activity against potentially pathogenic microorganisms [39].

The results obtained in the conducted study showed that all viable and heat-killed isolates have comparable auto-aggregation ability ranging from 17.08 to 48.87% with the best results obtained after 24 h of incubation. Many other studies observed the increase of auto-aggregation ability of LAB cells proportionally to time, with the highest values after 4 h [46], 5 h [38,54,55], and 24 h of incubation [45,51,52]. Overall, in the present study, the samples with cells exposed to heat stress for 5 and 15 min showed higher auto-aggregation ability compared to viable cells. Similarly, Shin et al. [56] observed that the auto-aggregation values of heat-killed CBT LA1 cells rose by 32.51% compared to viable cells (*p* < 0.05).

Since moderately heat treated cells appeared to have higher auto-aggregation ability than viable cells, these properties are likely to contribute to the higher adhesion rate to intestinal epithelial cells for colonization. Consequently, tested in the present study heat-killed cells of LAB strains or postbiotics from these bacterial cells can probably exert beneficial effects such as modulation of the immune response and development of a gut barrier against pathogens to a similar or greater extent than viable cells. However, this thesis should be verified during further studies.

### 4.3. Co-Aggregation Assays with Pathogens

Co-aggregation is a process by which genetically different bacterial cells become attached via specific molecules forming a complex of multi-species biofilms [63]. The ability of LAB strains to co-aggregate with pathogens enables forming the effective barrier that prevents colonization of intestinal epithelium by pathogens [64].

In the conducted study, measurements of the number of bound tested LAB cells to pathogenic bacteria were made at the beginning of the experiment, after an hour, and after 24 h of incubation. Regarding co-aggregation with *L. monocytogenes* and *S. aureus*, the highest percentages of co-aggregation ability were obtained directly after combining the cells of both strains (LAB versus pathogen), with a sedimentation time of 0. It was shown that the ability to form co-aggregates by the tested strains was greater in the case of *L. monocytogenes* compared to *S. aureus*. In the previous study [65] according to the data obtained from the well-diffusion method, the highest antimicrobial activity of investigated LAB strains (including strains used in the present study) was observed against *L. monocytogenes* and the lowest against *S. aureus*. This study seems to maintain this relation.

It is difficult to estimate unequivocally that thermal inactivation improved the co-aggregation ability with *L. monocytogenes* and *S. aureus*. Several significant differences in favor of the heat-killed LAB cells were noted, but cannot be applied to all tested strains. It is also intractable to refer to the results of research by other authors since there are limited data available. Tareb et al. [66] examined the co-aggregation ability of viable and heat-killed cells of *L. rhamnosus* 3698 and *L. farciminis* 3699 with the feed/food chain pathogens: *Escherichia coli*, *Salmonella* spp., *Campylobacter* spp. and *Listeria monocytogenes* after 24 h of incubation at 20 °C. The ability of *Campylobacter* strains to co-aggregate with probiotic strains was significantly better than that of other pathogenic strains tested (*p* < 0.05). Additionally, irrespective of the lactobacilli’s physiological status (viable or killed), the coaggregation ability remained similar. On the other hand, in the present study, the strain-specific co-aggregation ability was found. Similarly, Kim et al. [67] indicated that *Lactobacillus acidophilus* 11869BP heat-killed cells in comparison to viable cells had a higher co-aggregation ability with *Salmonella* Typhimurium. LAB viable cells displayed 43.5% co-aggregation ability, while heat-killed LAB cells displayed 55% ability [67]. Additionally, Ding et al. [68] demonstrated that heat-killed *Lactobacillus acidophilus* was able to co-aggregate with *Fusobacterium nucleatum*. However, after 15 min and 30 min of incubation the viable cells bound to the *F. nucleatum* to a slightly greater extent than the heat-killed cells (*p* > 0.05) [68], which was in line with the present study. It can be concluded that co-aggregation ability may be related to both LAB, as well as pathogenic strain properties and their bi-directed interactions.

### 4.4. Inhibition of Pathogens Adhesion

The protective role of probiotic bacteria against pathogens in gastrointestinal conditions and the underlying mechanisms is the subject of scientific interest [69]. Adhesion to and colonization of the mucosal surfaces have been identified as protective mechanisms against pathogens [69]. In the present study, the ability of the LAB to inhibit the adhesion or to compete with pathogens was assessed using the mucin from the porcine stomach model. Results of inhibition of pathogens binding to mucin by competition with tested LAB strains demonstrated that probiotic cells were very effective in case of inhibiting *L. monocytogenes* adhesion, and the *S. aureus* adhesion inhibition by tested bacterial strains showed high variability. However, heat-killed cells (especially 15 min) were more efficient than viable cells. Whatever the mechanism of exclusion, both heat-killed and viable cells of LAB strains prevented efficiently the adhesion of the pathogens to mucin.

In accordance with the results of conducted study, it has been previously reported that certain thermal inactivated cells of probiotics strains or postbiotics can inhibit mucosal or epithelial adhesion of different pathogens. Results of the study of Tareb et al. [66] demonstrated that when heat-killed cells were tested, the exclusion of *C. jejuni* CIP 70.2T was improved, especially for heat-killed forms of *L. farciminis* 3699 (from 15 to 70%). Also in our previous study [65] competition for adhesion to the human intestinal epithelial cell line Caco-2 of selected indicator strains (*E. coli*, *S. aureus*, *L. monocytogenes*, and *Salmonella*) with viable and heat-killed (80 °C per 20 min) cells of *Lactobacillus* strains were executed (including strains used in the conducted study). It was observed that inhibition of adhesion to Caco-2 cells was higher when cells of indicator strains were treated by viable than heat-killed cells of LAB. Although, in most samples, the effect did not differ significantly (*p* > 0.05). The inhibition effect was elevated in the case of *E. coli*, *Salmonella*, and *L. monocytogenes*, and demoted in the case of *S. aureus.* However, the results observed by other authors are ambiguous [50,68,70].

As adhesion inhibition is influenced by cell-surface properties, in the conducted study the question of whether or not the viability of bacteria is an essential criterion was considered. Inhibition adhesion of pathogens by heat-killed LAB cells in the present study has been demonstrated, so the elucidation of improvement of inhibition adhesion by heat-killed cells is desirable. Thermal inactivation of cells may lead to the rupture of their walls, thereby releasing cytoplasmic contents, like DNA, as well as components of their walls, such as peptidoglycans, lipoteichoic acids, or heat-labile pili. The released bacterial components (postbiotics) could be responsible for the adhesion inhibition of pathogens [71,72]. It has been also proposed that non-proteinaceous molecules like EPS (exopolysaccharides) may be overproduced by the bacterial cells as a protective barrier against heat before they die and this may explain why heat-killed probiotic cells were found to adhere more strongly [73,74].

## 5. Conclusions

The conducted study provided new information regarding the use of viable or heat-killed forms of selected LAB strains. The process of fixation with heating did not cause loss of the tested probiotic properties of LAB cells. It was observed that the modulation of these features was bidirectional, but did not result in their disappearance. Thermal inactivation for 5 and 15 min was a particularly prospective fixative, strengthening adhesion and aggregation properties of selected bacterial cells.

Although the strains used in this study were previously tested for their probiotic properties, however, these properties are related to the viability of the bacteria cells. Postbiotic metabolites obtained after heat inactivation of culture offer wider possibilities of using such bacteria. Therefore, in the view of provided data, it can be concluded that the postbiotics of tested LAB strains exhibited a good ability to reduce the adhesion in vitro of some pathogenic bacteria to porcine mucin. Moreover, the abilities of auto-aggregation, co-aggregation with pathogens, and cell surface hydrophobicity were rather strain-specific characteristics of these microorganisms. Due to their cell-surface properties and antimicrobial activity, especially their anti-listeria activity, the selected postbiotics of LAB strains isolated from foods can be used as possible protective agents. As a limitation of the study, still unclear mechanism of postbiotic action should be underline. Therefore, further studies are needed to explain the molecular basis of changes in bacterial features as a result of thermal inactivation.

The properties of the heat-killed probiotic cells to remain abilities of pathogen adhesion inhibition could enable those strains to be kept functional in unfavorable conditions. Future studies should focus on development and tested of the manufacturing process in purpose to offer more stable products characterized by identical properties such as original, probiotic formulations.

## Figures and Tables

**Figure 1 pathogens-11-01260-f001:**
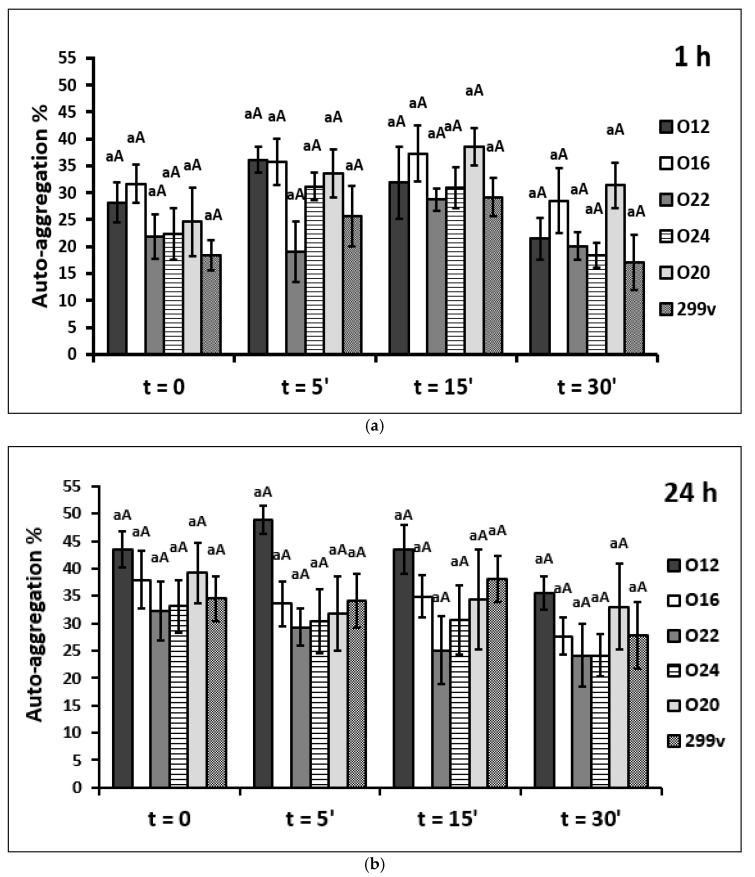
Influence of degree of thermal inactivation on tested LAB strains auto-aggregation percentage; (**a**) after an hour of incubation and (**b**) after 24h of incubation. Explanatory notes: The values are means of sixfold measurements with standard deviations. Means followed by different lowercase letters (a) between the different heat inactivation time at the same strain and different capital letters (A) between the same heat inactivation time at different strains are significantly different (*p* < 0.05) (Tukey’s range test). The names and symbols of the strains are indicated in the Material and Methods part of this paper.

**Figure 2 pathogens-11-01260-f002:**
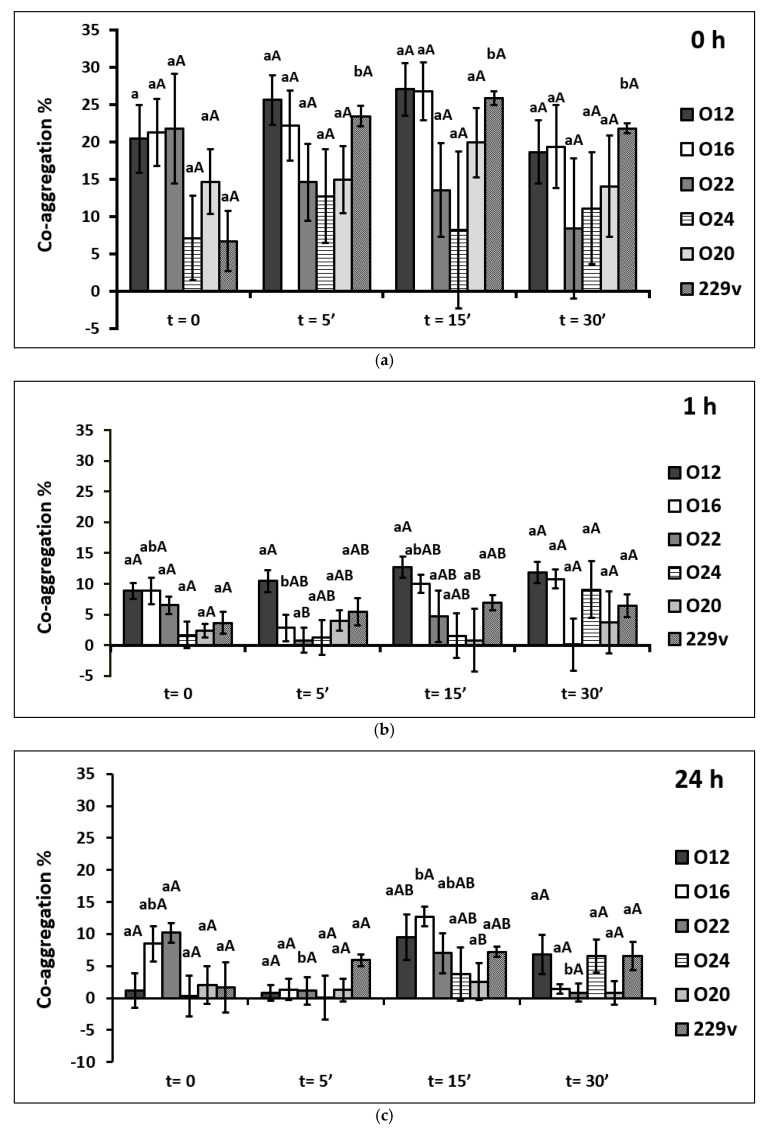
Influence of degree of thermal inactivation on tested LAB strains co-aggregation percentage with *Listeria monocytogenes*; (**a**) t = 0 h (**b**) after an hour of incubation and (**c**) after 24 h of incubation. Explanatory notes: The values are means of quadruple measurements with standard deviation. Means followed by different lowercase letters (a, b) between the different heat inactivation time at the same strain and different capital letters (A, B) between the same heat inactivation time at different strains are significantly different (*p* < 0.05) (Tukey’s range test or Kruskal–Wallis’s test). The names and symbols of the strains are indicated in the Material and Methods part of this paper.

**Figure 3 pathogens-11-01260-f003:**
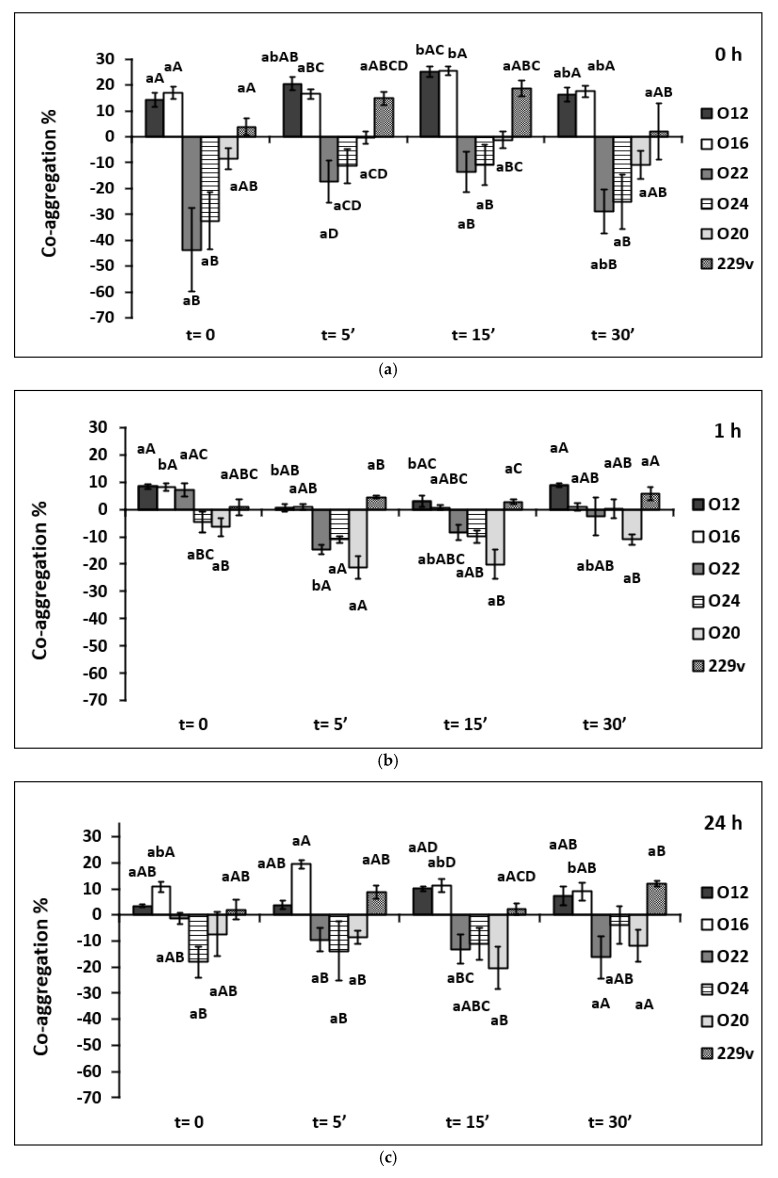
Influence of degree of thermal inactivation on tested LAB strains co-aggregation percentage with *Staphylococcus aureus*; (**a**) t = 0 h; (**b**) after an hour of incubation and (**c**) after 24 h of incubation. Explanatory notes: The values are means of quadruple measurements with standard deviation. Means followed by different lowercase letters (a, b) between the different heat inactivation time at the same strain and different capital letters (A, B C, D) between the same heat inactivation time at different strains are significantly different (*p* < 0.05) (Tukey’s range test or Kruskal–Wallis’s test). The names and symbols of the strains are indicated in the Material and Methods part of this paper.

**Table 1 pathogens-11-01260-t001:** Cell surface hydrophobicity of tested LAB strains against *p*-xylene and *n*-hexadecane.

Solvent	Heat Inactivation Time	Bacterial Strains						Heat Inactivation Time × Bacterial Strains
		O12	O16	O22	O24	O20	299v	
***p*-Xylene, %**	t = 0	58.33 **^aAD^** ± 2.93	55.30 **^aAB^** ± 3.76	54.20 **^aAB^** ± 3.29	50.60 **^aB^** ± 2.36	64.10 **^aD^** ± 1.41	36.03 **^aC^** ± 2.30	
	t = 5′	58.47 **^aA^** ± 5.71	64.07 **^abAB^** ± 1.39	58.90 **^abA^** ± 3.74	64.73 **^bAB^** ± 5.26	72.97 **^aB^** ± 5.45	43.27 **^bC^** ± 1.67	
	t = 15′	58.83 **^aA^** ± 3.10	67.27 **^bAB^** ± 7.09	59.70 **^abA^** ± 1.01	65.37 **^bAB^** ± 4.32	70.80 **^aB^** ± 3.40	39.53 **^abC^** ± 2.31	
	t = 30′	52.57 **^aAC^** ± 7.26	68.70 **^bB^** ± 0.69	62.70 **^bAB^** ± 1.68	64.67 **^bB^** ± 5.45	69.50 **^aB^** ± 3.48	42.33 **^bC^** ± 0.21	
**Two-way ANOVA, *p***	*******	*******						*****
***n*-Hexadecane, %**	t = 0	15.87 **^aA^** ± 3.87	12.87 **^aA^** ± 1.96	17.43 **^aA^** ± 5.36	16.13 **^aA^** ± 2.98	13.70 **^aA^** ± 2.30	16.73 **^aA^** ± 2.35	
	t = 5′	18.13 **^aAB^** ± 1.26	17.23 **^bAB^** ± 0.23	24.13 **^aB^** ± 3.39	21.70 **^aAB^** ± 1.40	15.57 **^aA^** ± 2.18	14.60 **^aA^** ± 5.98	
	t = 15′	16.53 **^aAC^** ± 1.99	15.03 **^abA^** ± 1.20	24.73 **^aB^** ± 3.18	23.30 **^aBC^** ± 5.30	11.00 **^aA^** ± 2.35	14.70 **^aA^** ± 0.96	
	t = 30′	16.13 **^aAC^** ± 2.31	15.53 **^abAC^** ± 0.40	24.80 **^aB^** ± 2.78	20.67 **^aAB^** ± 2.96	16.80 **^aAC^** ± 2.40	14.03 **^aC^** ± 0.91	
**Two-way ANOVA, *p***	*****	*******						**NS**

Explanatory notes: The values are means of triplicate measurements ± standard deviation. Means followed by different lowercase letters (a, b) between the different heat inactivation time at the same strain and different capital letters (A, B, C, D) between the same heat inactivation time at different strains are significantly different (*p* < 0.05) (Tukey’s range test); *p*, the significance of effects, heat inactivation time, bacterial strains, and heat inactivation time x bacterial strains; NS, not significant; * *p* < 0.05; *** *p* < 0.001. The names and symbols of the strains are indicated in the Material and Methods part of this paper.

**Table 2 pathogens-11-01260-t002:** Percentage of pathogenic bacteria cells reduced by tested LAB strains in competition for adhesion to porcine mucin conditions (in vitro).

Pathogen	Heat Inactivation Time	Bacterial Strains						Heat Inactivation Time × Bacterial Strains
		O12	O16	O22	O24	O20	299v	
***L. monocytogenes*, %**	t = 0	98.60 **^aA^** ± 0.33	99.11 **^aA^** ± 0.29	96.68 **^aB^** ± 1.23	95.59 **^aB^** ± 0.42	95.96 **^aB^** ± 0.56	99.39 **^aA^** ± 0.27	
	t = 5′	98.17 **^aA^** ± 0.59	98.78 **^aA^** ± 0.27	94.10 **^bB^** ± 0.30	98.24 **^bA^** ± 0.20	98.13 **^bA^** ± 0.30	98.84 **^aA^** ± 0.76	
	t = 15′	98.24 **^aA^** ± 1.07	98.97 **^aA^** ± 0.30	96.32 **^abB^** ± 0.77	99.05 **^bA^** ± 0.63	98.56 **^bA^** ± 0.34	99.18 **^aA^** ± 0.52	
	t = 30′	98.25 **^aA^** ± 0.63	98.86 **^aA^** ± 0.32	96.22 **^abB^** ± 1.61	98.74 **^bA^** ± 0.55	98.59 **^bA^** ± 0.42	98.86 **^aA^** ± 0.53	
**Two-way ANOVA, *p***	*******	*******						*******
***S. aureus*, %**	t = 0	87.68 **^aAB^** ± 8.95	95.17 **^aA^** ± 0.40	52.59 **^aB^** ± 1.18	86.66 **^aAB^** ± 4.20	88.75 **^aAB^** ± 1.08	93.13 **^aA^** ± 0.21	
	t = 5′	88.30 **^aAB^** ± 7.12	95.43 **^aA^** ± 0.30	56.23 **^aB^** ± 1.12	87.68 **^aAB^** ± 1.81	90.85 **^aAB^** ± 5.77	91.29 **^aA^** ± 12.10	
	t = 15′	95.24 **^aAB^** ± 0.75	99.13 **^bB^** ± 0.66	81.99 **^bA^** ± 6.66	97.70 **^bAB^** ± 0.72	91.25 **^aA^** ± 0.49	97.13 **^aAB^** ± 0.31	
	t= 30′	84.17 **^aAC^** ± 4.02	97.96 **^bB^** ± 1.67	80.29 **^bC^** ± 5.03	90.37 **^aAD^** ± 3.02	90.87 **^aABD^** ± 3.19	95.46 **^aBD^** ± 0.23	
**Two-way ANOVA, *p***	*******	*******						*******

Explanatory notes: The values are means of quadruple measurements ± standard deviation. Means followed by different lowercase letters (a, b) between the different heat inactivation time at the same strain and different capital letters (A, B, C, D) between the same heat inactivation time at different strains are significantly different (*p* < 0.05) (Tukey’s range test or Kruskal–Wallis’s test); *p*, the significance of effects, heat inactivation time, bacterial strains, and heat inactivation time × bacterial strains; *** *p* < 0.001. The names and symbols of the strains are indicated in the Material and Methods part of this paper.

## Data Availability

Data will be provided upon request.

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
