# Peer review of "The Influence of the Degree of Thermal Inactivation of Probiotic Lactic Acid Bacteria and Their Postbiotics on Aggregation and Adhesion Inhibition of Selected Pathogens"

_pathogens, 2022, doi:10.3390/pathogens11111260_

Round 1

Reviewer 1 Report

This article evaluates the effect of thermal inactivation of potentially probiotic lactobacilli strains on their ability to compete with pathogenic microorganisms. Overall, it is a well-planned study that highlights that heat inactivation can preserve some of the advantageous characteristics of these lactobacilli. Although the results of this study are encouraging, some of the conclusions can be misleading. The article should be revised in light of the following comments. 

1. Title: title seems misleading after going through the whole article. throughout the article, the authors seem to be confused about whether it is about the probiotic effect of these specific strains or the article is about the heat-killed paraprobiotics or postbiotics.

2. Introduction: it is focused again on probiotics, their development, and their characterization. It should focus more on post-biotic development and its effects. Authors should provide a thorough review of paraprobiotics development and their comparison with probiotics. Also, add the advantages and disadvantages of paraprobiotics over probiotics in this section.

3. Since the strains used in this study were previously characterized as potential probiotics, why use these as paraprobiotics?

4. Were these strains previously not studied for their activity and inhibition of S. aureus and L. monocytogenes?

5. In my opinion, the information derived in this article is useful for future paraprobiotics development, therefore authors should focus on that aspect in the manuscript and conclusion, there is too much regarding their effects of living strains as well.

6. As per the title focus on the inactivated cell properties and also give limitations of the study. 

7. Rearranging the overall story and tone down the conclusions. 

Res

Reviewer 2 Report

The work presented in the paper was undertaken to investigate lactic acid bacteria strains for several properties important for their health-beneficial effects.

Generally, data are clearly presented, and the material and methods are thoroughly described. Nevertheless, after reading the manuscript, I have several issues that I believe the authors should address.

 General comments

 The title says, “The influence of the degree of thermal inactivation of Lactobacillus bacteria on aggregation and adhesion inhibition of selected pathogens”. This implies that the authors studied bacterial strains, members of the genus Lactobacillus. However, as mentioned at the very beginning of the Material and Method section, none of those 5 bacterial strains belongs to the genus Lactobacillus. Therefore, the title should be changed to avoid misleading. I may suggest using the general term “lactic acid bacteria” instead of Lactobacillus.

As the same name is used throughout the manuscript, the authors are advised to check carefully and correct this in the main text as well. I acknowledge that, for historical reasons, Lactobacillus is a familiar name to researchers from different areas, and probably not all are aware that in 2020 this genus was reclassified into 25 genera, including those used in this study Lacticaseibacillus and Levilactobacillus. Therefore, it could be useful to provide this information in the introduction.

The term “paraprobiotics” is only one of several terms used to describe non-viable cells that have proven beneficial health effects on the host. Recently ISAPP proposed (Salminen et al., Nature Reviews Gastroenterology & Hepatology. 2021 Sep;18(9):649-67) following the definition of postbiotic “preparation of inanimate microorganisms and/or their components that confer a health benefit on the host”. I agree with ISAPP that “the field would benefit from coalescing around the use of a single, well-defined and understood term rather than the use of disparate terms for similar concepts”.

I found the discussion to be too long and have many undue details. For instance, the authors discuss in very detail the molecular composition of bacterial cell walls that may attribute to the hydrophobicity, while none of such studies was conducted and presented in the current manuscript. The same applies to the discussion of aggregation and pathogen adhesion to mucin. The author conducted a basic assay for the estimation of cell adhesion to mucin, but again discussed over one page the molecular background of adhesion inhibition, involvement of EPS and its overproduction as well as mice models. To summarize, the current discussion is disproportional to the results presented in the current manuscript and, therefore, I would recommend abbreviating some parts.

Minor comments:

Tables 1 and 2.

The tables in their current form are difficult to read. Could it be possible to rearrange the table to reduce the number of columns and format letters showing significant differences as superscripts? As an example, the authors may reduce the number of columns from 11 to 8, making the table easier to read.

Fig.1.  Please check the letters. Currently, all are the same, even when there is a significant difference between the means.

Page 7, line 194: Change treats to treatments
